# Streptavidin-Saporin: Converting Biotinylated Materials into Targeted Toxins

**DOI:** 10.3390/toxins15030181

**Published:** 2023-02-27

**Authors:** Leonardo R. Ancheta, Patrick A. Shramm, Raschel Bouajram, Denise Higgins, Douglas A. Lappi

**Affiliations:** Advanced Targeting Systems, Inc., Carlsbad, CA 92011, USA

**Keywords:** saporin, immunotoxins, Streptavidin-ZAP, biotinylation, molecular surgery, immunolesioning, secondary conjugate, engraftment, cancer research, animal models, behavioral studies, targeted toxin

## Abstract

Streptavidin-Saporin can be considered a type of ‘secondary’ targeted toxin. The scientific community has taken advantage of this conjugate in clever and fruitful ways using many kinds of biotinylated targeting agents to send saporin into a cell selected for elimination. Saporin is a ribosome-inactivating protein that causes inhibition of protein synthesis and cell death when delivered inside a cell. Streptavidin-Saporin, mixed with biotinylated molecules to cell surface markers, results in powerful conjugates that are used both in vitro and in vivo for behavior and disease research. Streptavidin-Saporin harnesses the ‘Molecular Surgery’ capability of saporin, creating a modular arsenal of targeted toxins used in applications ranging from the screening of potential therapeutics to behavioral studies and animal models. The reagent has become a well-published and validated resource in academia and industry. The ease of use and diverse functionality of Streptavidin-Saporin continues to have a significant impact on the life science industry.

## 1. Introduction of Streptavidin-Saporin: Trade Name—Streptavidin-ZAP

Saporin is a Type I ribosome-inactivating protein (RIP) from the seeds of the plant *Saponaria officinalis*. Saporin is an N-glycohydrolase and removes a single base in the rRNA, A4324. This eliminates the ability of the ribosome to perform protein synthesis [1,2]. Because of the lack of protein synthesis, any affected cell will die. However, Type I RIPs do not have the cell-binding capacity that many of the ‘true’ toxins have. For example, ricin and abrin are Type II RIPs that have internalizing subunit proteins for cell entry. Abrin was made famous in the 1980 film “Blue Lagoon” as a way to commit suicide (called ‘sleep berries’ in the film).

Key to the cytotoxicity of saporin is its internalization into the cell via endocytosis. In general, endocytosis is the internalization of various cellular macromolecules, fluids, and solutes. Once the material is internalized into an early endosome, the vesicle matures into a late endosome before finally fusing with lysosomes. This fusion of the late endosome and lysosome, termed an endo-lysosome, is where the contents are degraded by lysosomal enzymes [3]. The science regarding the escape of saporin from the late endosome has been a topic of interest, and studies have examined this aspect of the endocytic process to quantify the endosomal escape of saporin into the cytosol of a targeted cell [4]. Additional studies address how a rate-limiting step of delivery of cellular payloads into the cytosol of cells can be due to poor endosomal escape and how additional research is being done to examine various chemical or genetic strategies that can assist in saporin’s escape from the endosome [5,6]. When investigating the details after endocytosis, Vago et al. compared saporin and ricin A chain along with other bacterial toxins, such as cholera toxin and Pseudomonas exotoxin, and studied their different intracellular routes to enter the cytosol of intoxicated cells [7]. In their study, they discussed how after endocytosis of ricin in mammalian cells, it undergoes retrograde transport through the Golgi complex to reach the endoplasmic reticulum. However, when saporin was challenged with the Golgi-disrupting drug Brefeldin A or the proteasome inhibitor chloroquine, cell toxicity was not affected. This indicated that saporin follows a pathway not dependent on the Golgi complex. These results were verified in the same publication by tracking fluorescently-labeled saporin colocalized with a late endosome/lysosome marker; no Golgi localization could be detected [7].

The toughness of saporin was illustrated in Stirpe’s original description of the molecule, and the various means of causing apoptosis are discussed [8]. For saporin to be cytotoxic, it must be conjugated to a molecule that binds to something on the cell surface and internalizes, taking saporin inside the cell with it. The first commercially available saporin conjugate specifically targeted cells that expressed nerve growth factor receptor (NGFR) 192-IgG-SAP, which is a conjugate of the rat-specific monoclonal antibody to the NGFR (192 IgG) and saporin [9]. Scientists quickly understood the power of specific cell targeting and asked for new conjugates that would help them understand the functionality of different cell types in different species.

In 1997 the first ‘secondary conjugate’ was released, an affinity-purified goat anti-mouse IgG conjugated to saporin that recognizes a primary mouse monoclonal IgG antibody. Once the components are mixed, the complex can be used in vitro to screen antibodies for internalization. It is called Mab-ZAP: Mab for the ability to recognize a mouse antibody and ZAP to differentiate it as a new line of saporin products [10]. A saporin (SAP) conjugate is targeted with a molecule recognized by an extracellular marker. A ZAP conjugate is not targeted. The secondary antibody must have a primary antibody targeting agent; in the case of Mab-ZAP, a mouse antibody to an extracellular marker. The ZAP product line was expanded to include targeting antibodies in several species (e.g., mouse, human, rat, alpaca) and antibody isotypes (e.g., IgG, IgM, IgY). These secondary immunotoxins are powerful in vitro tools to determine if a primary antibody binds and internalizes to the cell of interest. The need to screen antibodies for internalization is clear, and the ZAP technique is employed in laboratories worldwide to determine if an antibody properly recognizes a cell surface molecule on a particular cell type. It was not yet appreciated that once an antibody or other molecule is bound to a cell-surface protein, it will almost always be internalized. When the molecule is attached to saporin and is internalized, the cell will die. If the cell does not die, then the cell-surface binding is not present, or the binding is not sufficient.

In 2004, a conjugate of streptavidin and saporin was commercialized. Streptavidin-ZAP can be mixed with any extracellular targeting agent (e.g., antibody, peptide, cytokine, growth factor, aptamer) that is labeled with biotin and is stable enough to be used in vivo [11]. The imaginations and ingenuity of scientists around the world continue to reveal the plethora of applications for Streptavidin-ZAP.

### 1.1. History of Avidin/Streptavidin Conjugates

Since 1994 when Advanced Targeting Systems (ATS) was founded, it has always been a goal to live up to the title of the company by providing new methods of targeting specific cell types and delivery of interesting molecules inside them. The ATS catalog contains dozens of conjugates that enable targeted delivery, almost exclusively delivery of saporin.

The ‘Molecular Surgery’ technique [12] requires a targeting agent to specifically bind to a cell-surface molecule on the appropriate cell population. Specificity is usually provided by the targeting agent to a target. For instance, a peptide such as substance P only binds at high affinity to its receptor, which exhibits itself on the cell surface [13]. Given the number of biomolecules that bind cell-surface molecules, there is a treasure trove of options to utilize the technology.

One of the most fruitful discoveries in cell targeting was that cell-surface molecules could also be bound by antibodies that are made to cross-react specifically to them. When monoclonal antibodies arrived, a whole new roster of molecules that bind to specific cell types arrived, and a Nobel Prize was awarded [14].

In 1991, a new method of nerve cell analysis was described and termed ‘immunolesioning’. This method used a bio-affinity (often an antibody) targeting agent that would lesion specific populations of brain cells. Lesions have a great history in the field of neuroscience. An early example was described by M. Paul Broca in 1861 [15], in which he discussed language and how it is processed in the brain. He repeated the idea (already described by Jean-Baptiste Bouillaud [16]) that “in the analysis of a large number of clinical facts, followed by autopsies, it was concluded that there are somewhere, in these lobes, one or more convolutions that hold under their dependence one of the essential elements of the complex phenomenon of speech”.

Immunolesioning was a term devised by Ronald G. Wiley to describe the use of antibodies as targeting agents to deliver saporin. The first of these brain tissue-targeted molecules was 192-IgG-Saporin, an ‘immunotoxin’ utilizing the antibody 192-IgG that targets the low-affinity neurotrophin receptor, p75 [9]. The p75 receptor is expressed on the surface of cholinergic neurons of the basal forebrain, a group of neurons that are believed to be important in several functions and are cells that deteriorate in Alzheimer’s Disease (AD). As such, a toxin that removed these cells from the rat basal forebrain provided an excellent model for some of the facets of AD. It is a validated model for the disease [17,18,19,20,21,22]. The use of an antibody to a cell-surface brain marker for lesioning was firmly established in a series of papers [23,24,25,26].

Over the course of time, more targeted toxins were created with the conjugation of saporin to antibodies. ATS was established to meet and augment the demand for targeted toxins. In the beginning, the market consisted mostly of neuroscientists who wanted to study specifically-lesioned neurons; these immunotoxins were appreciated and created new techniques, new careers, and new discoveries [12].

The cost of synthesis and determination of efficacy was high in terms of money and human power. Help arrived from Texas with Ellen Vitetta, a renowned immunologist who worked on targeted toxins for removing cancer cells from blood [27]. Her group used the antibody-binding protein from Staphylococcus, Protein A, to individually bind to antibodies to examine their ability to eliminate a target cell type. The technique received some interest, but it only became widely used when some changes were made. ATS changed the idea slightly by using a strong-binding streptavidin covalently attached to saporin to couple with a biotinylated targeting antibody. By offering Streptavidin-ZAP commercially, the need for institutional transfer agreements was avoided, and scientists were free to use the secondary conjugate for research without restriction.

In 1940, biotin was discovered by several scientists who realized, in their research on chicken eggs, that they were dealing with something in common and agreed to name it biotin [28]. The next year Esmond Emerson Snell published research about a molecule that was limiting the amount of biotin in chick eggs. In 1941, Gyorgy and Snell published that the responsible protein was characterized and later named avidin [29]. A similar molecule, streptavidin, was isolated from a genus of Streptococcus. While Streptococcus is quite different from chickens, and despite only a 33% similarity in amino acid homology, the two had crystallographic similarities and a similar disassociation constant, the strongest binding in biology. An overlay of the two crystal structures shows an amazing similarity in protein structures from species in different classes [30]. Streptavidin was chosen over avidin as the moiety to be attached to saporin due to the lack of glycosylation and lower isoelectric point, which results in a lesser degree of nonspecific binding [31].

### 1.2. Modular Way of Screening Targeting Agents: In Vitro and In Vivo

In an earlier publication, we described saporin, a tool from the plant kingdom, and its uses and impacts as a commercial reagent [12]. A branch from that tree of information focused on modular tools known as secondary antibody conjugates. These conjugates were able to fulfill the goal of a more modular and efficient way of creating an immunotoxin. As with all product development, a few issues remained and were hindering factors that included: (1) the limitation of using an antibody as a targeting agent and (2) instability in vivo.

One option to answer these issues came in the form of streptavidinylated saporin (Streptavidin-ZAP), where the Biotin–Streptavidin reaction was considered the chemistry best suited for a modular conjugation technique. This reaction has been widely studied and described and has proven itself to be an appropriate choice for a modular way of screening targeting agents. It was largely seen as an advantage to have one molecule that can be utilized both in vitro [32,33,34,35,36,37,38,39,40] and in vivo [41,42,43,44,45,46,47]. This modularity provided researchers the advantage of developing and planning larger-scale projects from preliminary in vitro beginnings to late-stage in vivo behavioral studies, all using the same verified reagent. Unlike the limitation of antibody-drug conjugates (ADCs: antibodies conjugated to a drug payload), any molecule recognized on the cell surface can deliver saporin, e.g., peptides, growth factors, cytokines, aptamers. This broader targeting capability has applications in many diverse fields of research.

### 1.3. Chemistry and Structure of Streptavidin-ZAP

With the attachment of streptavidin to saporin, scientists in both academic and industry who were looking to either screen new potential therapeutics, purify cell populations, or study behavior after depletion of cells could do so by harnessing the strongest known noncovalent biological interaction: the bond between biotin and streptavidin (Kd = 4 × 10^−14^ M) [48]. See Figure 1 for the crystal structures of the separate domains of the Streptavidin-ZAP complex, saporin and streptavidin.

As expected, not all has been perfect with the Streptavidin-ZAP technology. One of the early issues noticed, and still is a common concern, is not the Streptavidin-ZAP portion of the technology but the composition of the biotinylated targeting agent. There are biotin-tagged molecules that are commercially available, but the binding site of the molecule could be compromised, or there may be additives that prohibit their use with Streptavidin-ZAP to make a targeted toxin. Kits are also available in many orthogonal versions that cater to various chemistry scenarios, which range from, but are not limited to, succinimidyl ester-activated sites to react with primary amines or maleimide-activated sites to react with thiol groups, where even the solubility of the biotin, aqueous or non-aqueous, can be chosen (Figure 2).

A crucial aspect in choosing an appropriate biotinylation method is the length of the linker between biotin and the targeting agent. The relatively large size and multiple primary amines or thiol groups make the biotinylation of an antibody convenient. However, with smaller targeting agents such as peptides, the available reaction sites are limited, as well as the added concern of inadvertently affecting the downstream ability of the molecule to interact with its cognate receptor. Choosing the correct size and method of linkage for peptides requires knowledge of the amino acids that are needed to react with the receptor so as not to interfere with that specific region and to avoid inhibition of downstream efficacy. With the idea that there will be multiple biotins on a given molecule, any steric hindrance caused by biotin may not only interfere with the biotin reacting with streptavidin but with the targeting agent reacting with a receptor. Choosing a longer spacer arm can increase the probability of more biotin molecules reacting with streptavidin without causing interference for the delivery of the payload, in this case, saporin.

Peptides, as well as lectins [35,36,51,52,53,54,55,56], are included in the list of successful targeting agents that can be biotinylated and mixed with Streptavidin-ZAP. With small peptides (especially ones lacking amino acids targeted as the usual binding reactive sites-primary amines or thiols), introducing amino acids into the sequence, or attempting site-specific conjugation at the N-terminus [57] may be appropriate. Table 1 lists the amount of biotinylated targeting agent needed to react equimolar with 25 micrograms of Streptavidin-ZAP. An important aspect of determining when Streptavidin-ZAP is an appropriate method for your screening or behavioral studies is to first calculate the amount you will need. Antibodies have been primarily the targeting agent of choice [32,34,38,39,40,43,44,45,46,47,56,58,59,60,61,62,63,64,65,66,67], and conveniently, due to the similarity in molecular weight between whole IgG (~150 kDa) and Streptavidin-ZAP (~136 kDa), a small amount (25 μg) of Streptavidin-ZAP was found to facilitate numerous experiments. It is important to note when the biotinylated targeting agent is smaller in size, specifically in studies done with peptides in the range of 2–6 kDa, the amount of equimolar Streptavidin-ZAP needed is also small (0.37–1.1 μg).

An example of a smaller targeting agent that has been used successfully with Streptavidin-ZAP is an aptamer: short sequences of synthesized RNA or DNA that can bind to specific target molecules [68,69]. Of the various therapeutic payloads available, small interfering RNA (siRNA) has been under investigation in the clinic as a potential drug. However, the delivery of the molecule is still considered a crucial aspect that needs to be overcome [69,70,71,72]. One method that has been adapted for the specific delivery of siRNA is to employ the use of aptamers [69,73,74,75,76,77,78,79,80,81]. Since aptamers are synthesized, the addition of a biotin molecule can be introduced into the nucleic acid sequence at either the 5′ or 3′ end or even internally in the sequence, depending on where the reactive site is located, and then reacted with Streptavidin-ZAP at a 1:1 molar ratio [69].

## 2. Behavior, Disease, and Animal Models

Advances in gene therapy have been numerous, as evidenced by approved therapies in the past decade. As of the date of this writing, there are 3805 studies listed on ClinicalTrials.gov that are testing gene therapies for various diseases and behavioral conditions [82,83]. There are still many complications and hurdles that gene therapies need to overcome in order to become more clinically prevalent. A largely researched procedure relating to the gene therapy field is hematopoietic stem-cell transplantation (HSCT). The first person considered to be cured of HIV/AIDS underwent repeated allogeneic (donor) stem cell transplantation [84], but this required a very debilitating ablation of existing bone marrow and was a unique treatment case. HSCT is a risky procedure and can lead to different complications [85,86], which is why it is only reserved for life-threatening diseases. It is also a growing hospital procedure, but still underutilized for a broader range of diseases due to mortality rates.

### 2.1. Conditioning Regime for Engraftment

The use of reagents such as antibodies to rid unwanted cell populations and to refine gene therapy was reviewed by Logan et al. [87]. Since then, further advances to increase the efficacy of this conditioning regime have utilized immunotoxins created with Streptavidin-ZAP. A 2018 review publication summarized the targeted conditioning studies at that point in time [41]. Griffen et al. and Van Hentenryck et al. have both summarized the current status of antibody-based drugs in pre-transplant conditioning regimes [47,88]. Saporin conjugates that are discussed in their reviews are Anti-CD45-SAP and Anti-CD117-SAP; both were originally created for pre-transplant conditioning utilizing Streptavidin-ZAP [43,89]. Palchaudhuri et al. used Streptavidin-ZAP to make Anti-CD45-SAP, which enabled efficient engraftment of donor cells and full correction of a sickle-cell anemia model and showed a major contrast with previous irradiation methods which entail many negative side effects (neutropenia, anemia, toxicity, etc.) [89]. Other models, such as humanized X-linked severe combined immunodeficiency (SCID-X1) mice, were treated with a single dose of Anti-CD45-SAP to selectively deplete cells with great efficacy [90]. Another Anti-CD45-SAP publication showed potential for treatment for RAG (Recombination activating genes) deficiency–an autosomal recessive disease that produces immunodeficiency [42]. Bone marrow chimeric mice generated using Anti-CD45-SAP have been utilized to confirm the presence of myeloid progenitors at the meningeal border of the brain, and to lay a foundation to unravel possible functions in central nervous system (CNS) surveillance and local immune cell production [91].

The use of Anti-CD45-SAP has been postulated as a potential alternative to myeloablative conditioning for the treatment of a wide range of malignant and non-malignant diseases, such as autosomal recessive osteoporosis and hemophilia [45,58]. Persaud et al. showed that the use of this conjugate, combined with pharmacologic Janus kinase 1/2 (JAK1/2) inhibition, enables major histocompatibility complex (MHC)-mismatched allogeneic HSCT [46]. There is great potential for HSCT when using saporin conjugates in a pre-transplant conditioning regime; this research was, in large part, enabled by the availability of Streptavidin-ZAP.

Anti-CD117-SAP, prepared using Streptavidin-ZAP and developed by Czechowicz et al., showed that a single dose led to >99% depletion of host HSCs, enabling rapid and efficient donor hematopoietic cell engraftment [43]. The authors also showed that there were no clinically-significant side effects, suggesting that using Anti-CD117-SAP could create a ‘non-myeloablative’ conditioning strategy. Attempting skin allografts with this conjugate resulted in an interesting study that showed the potential for patients without MHC-matched donors to be eligible for transplant when using Anti-CD117-SAP [92]. This conjugate has also been employed in a hemophilia A mouse model, where using the conjugate and an immunosuppression approach in HSCT seemed to correct for hemophilia A [93].

### 2.2. Cancer Research Applications and Target Screening

The identification of tumor-specific receptors has been difficult because most are also expressed in normal tissues. Streptavidin-ZAP has been frequently used in therapeutic cancer screening as a tool to screen and identify targeting agents to eliminate cancer cells or decrease tumors. The ongoing need to find a cure for cancer and/or to get better survival rates inspires researchers to develop therapeutics/drugs that eliminate new targets or provide a substitute for current methods/drugs.

Mesothelioma: is a type of cancer that occurs in the thin layer of tissue that covers many internal organs, e.g., lungs, stomach, and heart (mesothelium). It has long been associated with asbestos exposure [94]. According to the World Health Organization, the global incidence of mesothelioma in 2020 was 21,560 (men) and 9310 (women). Mortality figures were 18,681 (men) and 7597 (women) [95].

A novel mesothelioma cell surface antigen, ALPPL2, was identified that could be specifically targeted by the antibody M25. Streptavidin-ZAP conjugated to M25 IgG1 was used in an immunotoxin assay to assess functional internalization and demonstrate potency and specificity against epithelioid and sarcomatoid mesothelioma [38].To determine whether an antibody can be developed as an immunotherapeutic drug, binding and internalization need to be proven. Berhani et al. used Streptavidin-ZAP in conjunction with a uniquely developed anti-human NKp46 monoclonal antibody to investigate the activity of human NKp46, a Natural Killer (NK) activating receptor, and its role in NK cell biology [32]. Streptavidin-ZAP was conjugated to the biotinylated human NKp46 antibody and tested on activated NK cells and an NK tumor cell line, which resulted in a decrease of cells and inhibition of cell growth, respectively. Therefore, the human NKp46 antibody can potentially be used to develop a drug to treat NKp46-dependent diseases like Type I diabetes and NK- and T-cell-related malignancies. This is just one example of the high value of saporin conjugates in effective drug development.

Thyroid cancer: develops in the thyroid gland, the butterfly-shaped gland at the base of the neck. According to a multi-national study of global incidence rates of thyroid cancer, in 2020, there were 10 per 100,000 women and 3 per 100,000 men with thyroid cancer. Mortality rates were less than one per 100,000 in most countries and in both sexes [96].

Streptavidin-ZAP conjugated to biotinylated Chlorotoxin [97] (CTX-SAP) was used to selectively target Matrix Metallopeptidase 2 (MMP-2), which is known to be expressed by ML-1 thyroid cancer cells. Chlorotoxin inhibits the enzymatic activity of MMP-2 and therefore reduces the surface expression of MMP-2. The in vitro studies showed that CTX-SAP decreased the number of ML-1 thyroid cancer cells in a dose-dependent manner [37].

Breast cancer: is the most common type of cancer worldwide. Estimates for 2023 from the American Cancer Society predict that there will be 300,590 new cases (297,790 for women, 2800 for men) and 43,700 deaths (43,170 for women, 530 for men) due to breast cancer [98]. Many antibody-drug conjugates (ADCs) have been developed to deliver toxins to breast cancer cells. The most common ADC in clinical use is trastuzumab-emtansine (T-DM1) [99].

A novel ADC to target HER2-positive breast cancer was prepared by conjugating biotinylated trastuzumab to Streptavidin-ZAP (T-ZAP) [34]. The study showed that the use of T-ZAP in trastuzumab-resistant cells resulted in more cell killing compared to T-DM1. Therefore, T-ZAP might be used to overcome trastuzumab resistance. Another way to overcome trastuzumab resistance is using a photochemical internalization technology. See Section 2.3.Triple-negative breast cancer (TNBC) is characterized by tumors lacking HER2, estrogen receptor, and progesterone receptor. TNBC has proven to be very difficult to treat, in large part because of the absence of consensus targets on the surface of the tumor cells. Damelin et al. [65] empirically established a set of surface markers associated with TNBC tumor-initiating cells, as produced by patient-derived xenografts. Ephrin-A4, which is overexpressed in TNBC and ovarian cancer, was selected as a therapeutic target, and a cell line transfected with the ephrin-A4 gene was challenged with two versions of biotinylated anti-ephrin-A4 coupled to Streptavidin-ZAP. Both the mouse monoclonal and the humanized antibodies reached an EC50 of 10 ng/mL, indicating that ephrin-A4 has promise as a therapeutic target for TNBC.

Gastric cancer: stomach cancer, also known as gastric cancer, is generally classified as cardia (upper stomach) and non-cardia (lower stomach). It is the third leading cause of cancer-specific death worldwide with 1.1 million new cases and 770,000 deaths in 2020. Incidence rates were, on average, twice as high in males than in females [100].

A study published in 2018 [60] targeted the gastric adenocarcinoma cell line AGS with anti-CDH17, an antibody against the extracellular domain of Cadherin-17 (CDH17), which is expressed in gastric cancer. Since CDH17 is composed of seven extracellular cadherin domains, it was found that a cocktail in which immunotoxins recognize different epitopes on CDH17 resulted in additive/synergistic effects and, therefore, the best cytotoxic results.Streptavidin-ZAP was used to measure the internalization of antibodies against novel surface markers [40]. A novel single-chain variable fragment (scFv) 78 was tested against tumor endothelial marker 1 (TEM1). The scFv78 was evaluated as a tool for molecular imaging, immunotoxin-based therapy and nanotherapy. Streptavidin-ZAP was used to evaluate whether scFv78 can be used in vitro to deliver an immunotoxin selectively to TEM1-positive cells. The results showed dose-dependent cytotoxicity that was specific to TEM1-positive cells.

Ovarian cancer: a cancerous growth of cells that forms in the ovaries. Ovarian cancer in women is relatively rare in comparison to other cancers and represents ~1% of all new cancer cases in the U.S. The National Cancer Institute estimated that in 2022, there would be 19,880 new cases with 12,810 deaths [101].

Streptavidin-ZAP was used to evaluate therapies targeting human tumor vasculature and human cancer stem-like cells [102]. Targeting tumor vascular markers (TVM) is difficult since the vasculature expression profile of tumor types tends to be very different. Burgos-Ojeda et al. established a human embryonic stem-cell-derived teratoma and tested it as a model for TVM expression by challenging primary human mesenchymal stem cells (MSCs) in vitro. They also evaluated TVM expression in a human embryonic stem-cell-derived teratoma (hESCT) tumor model previously shown to have human vessels. The direct intravenous injection into subcutaneous tumors resulted in a temporary lack of tumor growth or regression of the tumor and tested the ability of the hESCT model to enhance the engraftment rate of primary human ovarian cancer stem-like cells.

Prostate cancer: one of the most common types of cancer in men. Prostate cancer in women is rare, affecting the glandular tissue below the bladder known as Skene’s glands [103]. In men, prostate cancer is the 2nd most commonly occurring cancer. There were more than 1.4 million new cases of prostate cancer in 2020 [104].

Toxicology studies of biotinylated anti-CD46 mixed with Streptavidin-ZAP were performed in non-human primates that showed the potential of CD46 for use as a target in adenocarcinoma and neuroendocrine types of metastatic castration-resistant prostate cancer (mCRPC) [61]. CD46 is a multifunctional protein that negatively regulates the innate immune response. Cytotoxicity data showed that the immunotoxin killed mCRPC cells but not control cell lines: a benign prostatic hyperplasia epithelial cell line and a primary normal human liver cell line that expressed low amounts of human CD46.Kuroda et al. examined the cytotoxic efficacy of anti-prostate-specific membrane antigen (PMSA) conjugated to saporin on PMSA-positive cell lines. hJ591, also known as rosopatamab, is a humanized anti-PMSA antibody that was biotinylated and combined with Streptavidin-ZAP [105]. The immunotoxin was specifically cytotoxic to PMSA-positive cells and exhibited anticancer activity in a xenograft model. This research demonstrates the anticancer potential of targeting PMSA.Wüstemann et al. [56] also investigated the use of a similar Anti-PMSA-Saporin conjugate. Binding potency was comparable to that of naked antibodies, and in vivo experiments proved potent for selective tumor growth inhibition in mice bearing lymph node carcinoma of prostate (LNCaP) tumors [56].

Multi-cancer targeting. Some studies using Streptavidin-ZAP targeted multiple cancers through shared characteristics.

*Ras-transformed cancers*. According to Cancer.gov, approximately one-third of cancers, including a high percentage of pancreatic, lung, and colorectal cancers, are the result of mutations in RAS genes [106]. Macropinocytosis, the internalization of large endocytic vesicles called macropinosomes, is upregulated in Ras-transformed cancers. Ha, et al. demonstrate the screening and validation of antibodies that utilize the macropinosome pathway [33]. One method used was to biotinylate the antibodies and combine them with Streptavidin-ZAP at a 1:1 M ratio. The conjugate was applied to cells in a concentration curve starting at 200 nM to demonstrate internalization and cell killing. The results showed receptor-dependent micropinocytosis that allows tumor-targeting antibodies to be internalized via the macropinocytosis pathway.*Breast and ovarian cancers*. An antibody (A19) was produced in mice using human embryonic stem cells (hESCs) as the immunogen. A19 binding studies revealed that this antibody recognizes the N-glycan epitope on Erb-b2 (Erb-b2 receptor tyrosine kinase 2) that is expressed by many different breast cancer and ovarian cancer cell lines [62]. Biotinylated A19 was mixed with Streptavidin-ZAP and tested in vivo in nude mice [62]. Each nude mouse was injected in the right flank, subcutaneously, with 5 × 10^6^ SKOV3 (a human ovarian cancer cell line) cells; the immunotoxin was administered intraperitoneally at 37.5 μg/dose. After 10 weeks, a 60% reduction in tumor size was observed, which indicates that A19-Saporin suppressed tumor growth.*Small cell lung cancer and neuroblastoma*. Streptavidin-ZAP was conjugated to a mouse anti-human-HuD monoclonal antibody to eliminate small cell lung cancer (SCLC) and neuroblastoma (NB) cells that express HuD, a neuronal RNA-binding protein [107]. The immunotoxin was tested in vitro and showed cytotoxicity at very low concentrations. After a killing baseline was established, Anti-HuD-Saporin was injected (1 mg/kg) directly into subcutaneous tumors generated in mice which resulted in a temporary lack of tumor growth or regression of the tumor. The results indicate the potential of HuD as a therapeutic target for SCLC and NB.

### 2.3. Photochemical Internalization

Definition: Photochemical internalization (PCI) is a drug and gene therapy delivery method that uses photosensitizers and light to facilitate the endosomal escape of the macromolecules, such as proteins and nucleic acids, into the cytosol [108].

Photosensitizers are placed in endocytic vesicles that lead to the rupture of the endocytic vesicles following light activation [109]. This technology is especially useful when Type I ribosome-inactivating proteins (RIPs) and other macromolecules that have no binding chain and cannot enter a cell on their own are being delivered. PCI is an important technology used in cancer research as targeted therapies can be trapped in the lysosome and compartmentalized away from the target. PCI increases the efficacy of those compounds by releasing the therapeutic portion of the molecule into the cytosol using light.

Usage of PCI in combination with Streptavidin-ZAP conjugates. Wong et al. showed light-controlled elimination of programmed death ligand-1 (PD-L1+) cancer cells and immunosuppressive cells in the tumor microenvironment by PCI of a PD-L1-targeting immunotoxin (Figure 3) [39]. PD-L1, also known as CD274 or B7-H1, is important for tumor progression and immune escape. It is crucial to eliminate PD-L1 because even though there are clinically used immune checkpoint inhibitors of PD-L1, immune checkpoint therapy is not effective in most patients [39]. In Wong et al.’s publication, the PD-L1+ triple-negative breast cancer MDA-MB-231 cell line was targeted by the immunotoxin Anti-PD-L1-Saporin via PCI and resulted in an in vitro proof-of-concept of PCI-enhanced targeting and elimination of PD-L1-positive immunosuppressive cells.

Chondroitin sulfate proteoglycan 4 (CSPG4/NG2) was used as a targeting agent as it is highly expressed in triple-negative breast cancer (TNBC) and malignant melanoma. CSPG4 is instrumental in tumor cell growth and survival and furthers chemo- and radiotherapy resistance [59]. CSPG4-specific mouse monoclonal antibody 225.28, an IgG2a, was biotinylated and mixed with Streptavidin-ZAP, then photochemically delivered to TNBC and melanoma cells. The cytotoxicity was highly dependent on the light dose and expression of CSPG4. This study validated the CSPG4-targeting concept in vitro, which built the foundation for future studies in which cancer cells can be eliminated in a specific and light-controlled way.

The work from Hamakubo et al. showed the importance of PCI as an alternative and enhanced way to increase the quality of life of patients that suffer from head and neck squamous cell carcinoma (HNSCC) [63]. Conventional treatments like surgery, chemotherapy and radiotherapy often result in long-term complications [110]. One goal is to reduce those complications by developing and using an antibody therapy. The axon guidance receptor, Robo1, was targeted with biotinylated anti-Robo1 mixed with Streptavidin-ZAP in HNSCC cell lines. It was shown that the use of the PCI technology improved the cytotoxic effects greatly in low-level receptor-expressing cells.

PCI has also proven to be a way to overcome certain obstacles such as trastuzumab resistance. Many patients develop acquired resistance to trastuzumab, the monoclonal antibody recognizing HER2 and used as a breast cancer therapeutic. One of the modes of resistance is that the therapeutic is trapped inside an endocytic vesicle. PCI facilitates cytosolic release. Therefore, Trastuzumab-Saporin (Biotinylated Trastuzumab mixed with Streptavidin-ZAP) was tested in combination with PCI using Amphinex, a photosensitizer approved for clinical use. It was shown that PCI increased the cytotoxicity of Trastuzumab-Saporin on Trastuzumab-resistant HER2(+) cells. This study also demonstrated the importance of administering the immunotoxin prior to light exposure [111].

Another example of light-triggered drug delivery is the research targeting the CD44 receptor, a common cancer stem cell (CSC) marker. It was targeted by a biotinylated pan CD44 monoclonal antibody bonded with Streptavidin-ZAP (IM7-Saporin). This resulted in the efficient and specific killing of several CD44-expressing cancer stem cells. This study demonstrated the efficacy of PCI in conjunction with targeted toxins to treat some cancers, especially when cancer cells are resistant to therapeutic agents [112]. Bostad et al. also targeted CD133, another CSC marker for several different cancers. The challenge with this marker is that CD133 is also expressed in non-cancer stem cells, so it was crucial to only deliver immunotoxins to CD133-positive cancer stem cells [113]. This was achieved by employing light-activated PCI technology to specifically deliver Anti-CD133-SAP to the target cells, which caused cytotoxicity in femtomolar concentrations. This technology uses light activation of tumor-accumulating photosensitizers located on the membrane of endocytic vesicles to deliver the immunotoxin.

The epithelial cell adhesion molecule (EpCAM) that is expressed in many human carcinomas and cancer stem cells is another promising tumor target. A novel human EpCAM-targeting monoclonal antibody, 3-17I, was developed and used in several assays to demonstrate the antibody’s potential as an oncology tool [114]. In one series of assays, EpCAM-positive cancer cells were treated with 3-17I-Saporin (biotinylated 3-171 mixed with Streptavidin-ZAP) and the photosensitizer TPCS2a (Amphinex), followed by light exposure. The immunotoxin was shown to have specific cytotoxicity on several different cancer cell lines over a range of concentrations during light exposure.

Targeting the epidermal growth factor receptor (EGFR) has become very important in cancer research and therapy due to its overexpression in many different types of solid tumors and its association with metastasis and poor prognosis [115]. This work, published by Yip in the Selbo laboratory, demonstrated an in vitro proof-of-concept for delivering cetuximab–saporin to EGFR-expressing cells by PCI. The targeted toxin consists of the biotinylated chimeric murine-human IgG1 monoclonal antibody, cetuximab, and Streptavidin-ZAP. The conjugate was applied to three different human cancer cell lines, showing increased specificity and toxicity against cells expressing the EGFR.

### 2.4. Immunology

One of the more common methods of using Streptavidin-ZAP is to couple the complex with biotinylated antibodies. However, there are also many instances of biotinylated ligands, peptides, and even biotinylated major histocompatibility complex (MHC) tetramers in immunology fields.

B-Cell Targeting. An example of utilizing Streptavidin-ZAP to study ligands that target CD22 was shown by Collins et al. [116]. CD22 is a potential target for immunotherapy of B-cell lymphomas. The authors examined the equilibrium between CD22 and the cis and trans forms of its ligands using high-affinity sialoside probes. They also demonstrated that a biotinylated probe specific for CD22, when combined with Streptavidin-ZAP, can eliminate several different lymphoma cell lines.

Rheumatoid arthritis is a chronic disease that is accompanied by anti-citrullinated protein antibodies (ACPA) produced by autoreactive B cells. A study in 2018 used a synthesized cyclic citrullinated peptide (CCP) antigen suitable for B-cell receptor binding and demonstrated that binding by ACPA was impaired upon manipulation of the residue [117]. The data were generated using biotinylated CCP mixed with Streptavidin-ZAP in cell viability assays. The results marked an important step towards antigen-selective B-cell targeting in general and, more specifically, in rheumatoid arthritis.

T-Cell Targeting. An example of using Streptavidin-ZAP to deplete specific T cells was published in 2010; Akiyosi et al. used the dendritic cell-associated heparan sulfate proteoglycan-dependent integrin ligand (DC-HIL) as the targeting agent. DC-HIL is exclusively associated with syndecan-4 (SD-4), which is expressed in some activated T cells [118]. A similar study was done with Sézary syndrome cells that overexpress syndecan-4 [119].

Hess et al. investigated whether pathogenic T cells could be depleted via Streptavidin-ZAP coupled to MHC class I tetramers to kill antigen-specific CD8+ T cells [120]. Their work showed the therapeutic potential for using cytotoxic tetramers to eliminate specific T cells. This same strategy was employed in vivo to delay diabetes in non-obese diabetic mice [55]. The Hess group also used biotinylated peptide-MHC class I tetramers with Streptavidin-ZAP to selectively deplete a population of alloreactive T cells in mice to determine that toxic tetramer administration prior to immunization increased survival of cognate peptide-pulsed cells in an in vivo cytotoxic T lymphocyte assay and reduced the frequency of corresponding T cells [52]. More research towards T-cell depletion utilizing Streptavidin-ZAP and biotinylated MHC tetramers came from Sims et al., where they found that following a significant transient depletion of cells, the population rebounded and reached a higher percentage of total CD8+ T cells than before the depletion. This research provides a further understanding of the ‘flexibility and turnover’ of these cells [66].

Other Applications. Outside of T cells and B cells, Streptavidin-ZAP has also been used to deplete dendritic cells (DC). Alonso et al. depleted inflammatory DCs with biotinylated anti-CD209 via intravenous injection in a mouse animal model of induced inflammatory DC formation [67]. The authors suggest that the depletion of inflammatory DCs could be useful in understanding inflammatory diseases such as psoriasis. Depletion of natural interferon-producing cells (IPCs) was demonstrated with an IPC-specific biotinylated antibody in vitro [121].

Facciabene et al. made immunotoxins with Streptavidin-ZAP to biotinylated antibodies for CCR10 and CCR3 (Anti-CC10-Saporin and Anti-CCR3-Saporin). They investigated whether a direct link between tumor hypoxia and tolerance occurs through the recruitment of regulatory cells [122]. Their findings showed that peripheral immune tolerance and angiogenesis programs are closely connected and cooperate to sustain tumors.

### 2.5. Neurosciences

Streptavidin-ZAP is also a tool for study in the neurosciences [64]. Although recent work has shown that some intrinsically photosensitive retinal ganglion cells (ipRGCs) are responsible for processing nonimage-forming visual functions, it is unclear whether the ipRGCs or conventional RGCs modulate affective behavior. Huang et al. injected 2 μg of Melanopsin-SAP, or biotinylated Anti-Cholera Toxin B-subunit coupled to Streptavidin-ZAP, into each eye of gerbils. The data suggest that retino-raphe signals modulate dorsal raphe nucleus serotonergic tone and affective behavior. Melanopsin-SAP is a chemical conjugate of an affinity-purified rabbit polyclonal antibody to mouse melanopsin and saporin that specifically eliminates ipRGCs.

### 2.6. Gastroenterology & Cardiac Function

Research related to gastroenterology and cardiac function has also utilized Streptavidin-ZAP. Rothenberg et al. published in the journal Gastroenterology regarding the identification of a cKit-positive (CD117-positive) secretory cell that supports Lgr5 (leucine-rich repeat-containing G-protein coupled receptor 5)-positive stem cells in mice [123]. The group showed that cKit-positive cells promote the organoid formation of Lgr5-positive cells. They did this by culturing isolated cKit-positive cells and Lgr5-positive cells together, which promoted organoid formation. When organoids were depleted of cKit-positive cells using the Streptavidin-ZAP-created Anti-CD117-SAP, organoid formation decreased.

Accidental injury to the cardiac conduction system (CCS) was recently studied using a biotinylated CCS-specific antibody bonded with Streptavidin-ZAP [44] to explore cardiac function. The CCS is a network of nodes, cells, and signals that control the heartbeat; injury to this network can be a complication in cardiac surgery. In this publication, Goodyer et al. engineered tools to target and visualize the CCS following a single intravenous injection in mice. They modulated cardiac function by using a biotinylated antibody to contactin-2 mixed with Streptavidin-ZAP to perturb the CCS. It is research such as this that paves the way for further understanding of cardiac function and lays the foundation for therapeutic development.

## 3. Conclusions

The streptavidin-biotin technology continues to evolve. Medical uses are beginning to emerge. For example, pretreatment with biotin-modified endothelial cells to avidin-functionalized stents provides a 30% reduction in-stent restenosis [124]. Human pharmaceutical applications using the streptavidin-biotin technology are more difficult due to biotin supplement interference with attempted interventions or treatments. Biotin supplements (vitamin B7) are commonly used to treat thinning hair or some types of rashes. While there is no scientific evidence that supplements work for these symptoms, the possible interference with some clinical laboratory tests is a concern.

Research using the streptavidin-biotin technology continues to undergo changes. In 2007, a review of genetically-engineered molecules was published that describes the differences in constructs and the benefits they provide [125]. It is important to pay attention to new developments and utilize improvements to conjugate components that will advance technology and accelerate scientific discovery. The popularity of the streptavidin-biotin system is not surprising due to the broad applications and areas that rely on the strong bond and stability it provides. Scientists are strongly encouraged to use the appropriate control conjugate in experiments to protect the validity and reliability of research done with Streptavidin-ZAP. The control is a saporin conjugate that has no method of entry into a cell except through bulk-phase endocytosis. Dose-ranging studies that compare control to targeted conjugate easily reveal when a dose is too high and will have off-target effects. The control should never cause cell death.

After this visit to the world of Streptavidin-ZAP and its uses, it is clear that there have been profound contributions to scientific research using the combination of these two technologies: the streptavidin-biotin bond, and the effective cell killing by saporin. It is expected that new, exciting additions to the use of Streptavidin-ZAP will continue to be revealed in the years to come.

## Figures and Tables

**Figure 1 toxins-15-00181-f001:**
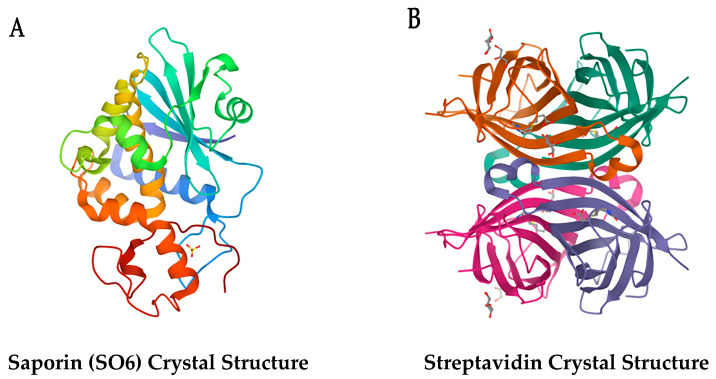
Crystal Structures of the separate domains of Streptavidin-ZAP. (**A**) Saporin MW: ~30 kDa [49]. (**B**) Streptavidin MW: ~55 kDa [50].

**Figure 2 toxins-15-00181-f002:**
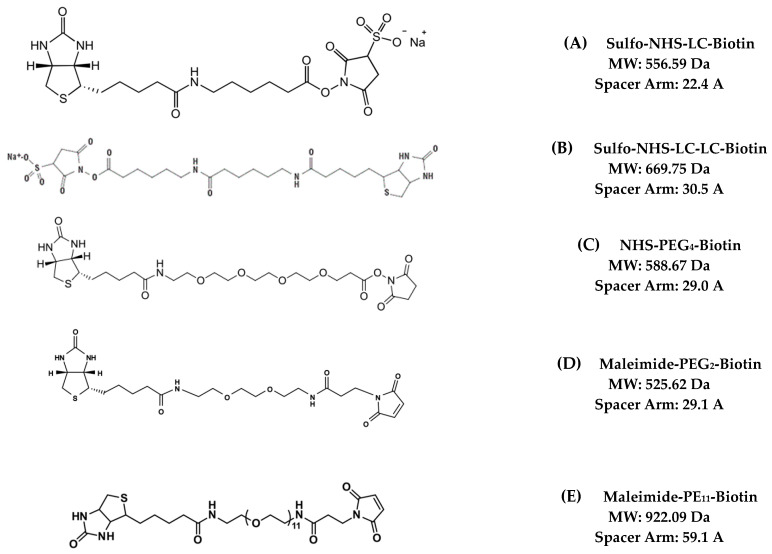
Molecular structure of some of the commonly used biotinylation strategies for antibodies.

**Figure 3 toxins-15-00181-f003:**
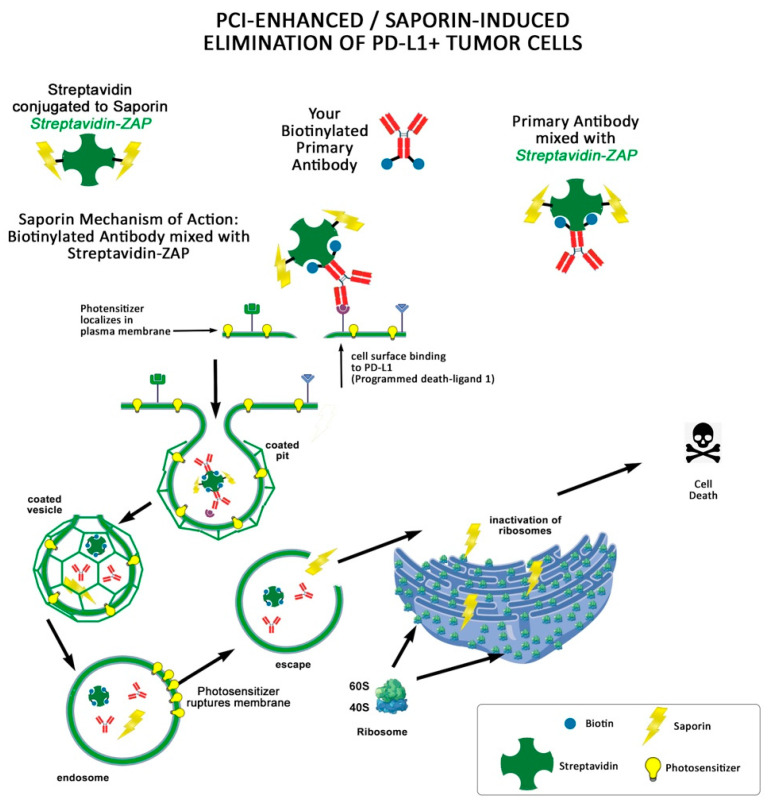
PCI-enhanced, saporin-induced elimination of PD-L1+ tumor cells. A photosensitizer is administered and localized in the cell membrane. Biotinylated Anti-PD-L1 mixed with Streptavidin-ZAP binds to its cell surface marker and is internalized. The cells are exposed to light that activates the internalized photosensitizer causing the membrane to rupture. This appears to intensify the effect of saporin to kill cells.

**Table 1 toxins-15-00181-t001:** Biotinylated targeting agents and corresponding Streptavidin-ZAP amounts needed to make a bonded conjugate.

Biotinylated Targeting Agent (TA)	Size of TA	Biotinylated TA Needed to React Equimolar with 25 μg Streptavidin-ZAP (MW: 136 kDa)
Antibody: Whole IgG	~150 kDa	29.41 μg
Antibody: F(ab’)2	110 kDa	20.22 μg
Antibody: F(ab)	55 kDa	10.11 μg
Antibody: single-chain variable fragment (scFv)	28 kDa	5.15 μg
Lectin (e.g., Isolectin B4)	28 kDa	5.15 μg
Growth Factor (e.g., Fibroblast Growth Factor)	16.5 kDa	3.03 μg
RNA Aptamers	13–17 kDa	2.4–3.1 μg
Peptides	6 kDa	1.1 μg (1100 ng)
Peptides	5 kDa	0.92 μg (920 ng)
Peptides	4 kDa	0.74 μg (740 ng)
Peptides	3 kDa	0.55 μg (550 ng)
Peptides	2 kDa	0.37 μg (370 ng)

## Data Availability

Not applicable.

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
