# Peer review of "Streptavidin-Saporin: Converting Biotinylated Materials into Targeted Toxins"

_toxins, 2023, doi:10.3390/toxins15030181_

Round 1

Reviewer 1 Report

The author reviewed a  toxin on Streptavidin-Saporin with very informative content. The manuscript was well-written and organized.

Only a few small issues should be addressed.

1.  Regarding the Structure of Streptavidin-ZAP, the figure legend is not sufficient to describe the structure. It would be better to label the name of the different domains in the complex.  Since you add the distance bar in the figure, then distance should be added.  While  PDB ID should be referred.  In line 134 the author states the bond between biotin and streptavidin (Ka = 1015 M-1). Ka means the rate constant. Here dissociation constant Kd should be used. The reference is missing here as well. 

2. In table 1, Check the font of the unit of peptides  amount  "ug" 

Reviewer 2 Report

I have read carefully the manuscript entitled: ‘Streptavidin-Saporin: Converting Biotinylated Materials into  Targeted Toxins’. The review is well written and convincing. In my opinion, the present version of the manuscript needs to be slightly revised before possible publication in the ‘toxins’ journal, as reported below.

- The authors used too many keywords, please reduce them.

-In introduction section, the authors should add information on the enzymatic activity of RIPs. Indeed, it is widely accepted that these enzymes possess N-β-glycosylase activity which can be highlighted be detection of β-fragment, upon aniline treatment. On this regard, a recent review could be helpful (i.e.: https://doi.org/10.3390/toxins14060403).

-In Figure 3 the authors clearly describe a eukaryotic cell (e.g. presence of RER). In this framework, I would like to remind them that eukaryotic ribosomes are made up of 60S and 40S subunits. please correct the apparent mistake.

Reviewer 3 Report

This manuscript is a review that focuses on the use of the  streptavidin-saporin conjugate and how it can be used along with biotinylated targeting agents to yield novel targeted toxins for the treatment of human disease. The manuscript is of significant interest to the field of targeted immunotoxins and is overall well written. However, key points are missing and the readability/flow of the manuscript could be improved by restructuring its content.  

Concerns:

1) The manuscript should start off with a brief introduction of what saporin is, where it comes from, its mechanism of action (its function), etc so that a general audience can understand the article. The authors should then introduce streptavidin and avidin and discuss how they are different and how they are the similar. The affinities of both SA and avidin should be given as part of this general introductory paragraph. The general concept of antibody-drug conjugates (ADC) should be introduced and how these biotin-SA conjugates are a unique form of ADC. A brief summary of the advantages of biotin-SA conjugates over conventional ADC would be a good addition and would help the reader understand the important of these conjugates. 

2) One important point not discussed in this manuscript is what happens once the conjugates are inside the cell. How is the strong bond of biotin and SA or avidin broken? or is it? How does the saporin toxin get to the cytoplasm? This may in large part be unknown, but the authors needs to discuss what is known. For example, one molecule of saporin has been shown to kill a cell so high concentrations aren't needed. 

3) The descriptions of each cancer type are oversimplified. More accurate and complete descriptions (current incidence and death rates, etc) would be beneficial. For example, prostate cancer is described as "one of the most common types of cancer." This is true for men, but not for women. Also, why do the authors stress that it can be slow growing and is confined to the prostate?" This can be true, but it can also be very aggressive and highly metastatic. Lastly, why is "multi-cancer targeting" in line 332-333 under prostate cancer? Seems like this should be a separate section.

4) Why haven't these conjugates moved towards clinical development? The immunogenicity of streptavidin (a bacterial protein) should be discussed. Are these conjugates really possible treatments for human disease? Some of these conjugates target proteins on normal cells. Trastuzumab for example. HER2 is expressed on cardiac cells. Delivery of a conjugate containing saporin would cause many more side-effects that trastuzumab alone. The problem with immunogenicity and off-target (side-effects) is very important and should be discussed. Any studies using Streptavidin-ZAP in monkeys or clinical trials (if any) should be emphasized. 

Minor concerns:

1) Many letters are capitalized that do not need to be. For example, saporin does not need to be capitalized unless it is the first word of a sentence or is part of a proper name. "Antibody-Drug Conjugate" also does not need to be capitalized. "Prostate-Specific Membrane Antigen" should be "prostate-specific membrane antigen."

2) Streptavidin-ZAP is mentioned out of the blue in the last line of the abstract. There is no context here or further information as to what this is. This reagent should be briefly explained so that the general reader knows what it is and where it comes from. 

3) In line 31 of the Introduction, the authors cite ref #1 (Lappi et al, 1985 BBRC 129:934-942) to refer to  the 192-IgG-SAP conjugate. However, this reference has nothing to do with this conjugate so this reference has not been cited properly. This reference could be cited in the introduction since it discusses the general properties of saporin, but shouldn't be cited when discussing the 192-IgG-SAP conjugate. 

4) References are often not provided. For example, in lines 79-80 a reference should be cited on "immunolesioning." The meaning of this term should also be given. Another example is in line 134, a reference should be provided for the Ka of biotin and streptavidin.

5) In line 97, the authors mention "the antibody-binding protein from Staphylococcus". Do the authors mean protein-A? If so, this should be stated. 

6) Figure 1 should have labels to denote where the 3 saporin molecules are and which one is streptavidin. 

7) The commonly used size of human IgG (1, 2, and 4) is 150 kDa, why do the authors list 160 in Table 1?

8) Section 2.2 is labelled "Cancer Therapeutic Screening." What about cancer therapy? Aren't any of these conjugates used for cancer therapy? If so this should also be discussed. 

9) In lines 262-265,  chlorotoxin is discussed, however, how this molecule targets the thyroid cancer cells is not clear. It seems like there are two toxins put together. This needs to be clarified. 

10) In line 344, what animal was used to produce the A19 antibody? The authors state that hESCs were used as the immunogen, but they do not mention the animal used. 

11) Figure 3 should not be referenced in the text in line 364. This paragraph is on general PCI. It should be referenced in the next section when PCI usage in combination with strepativin-ZAP is discussed. 

12) Figure 3 should be made as a general figure, meaning PD-L1 should be removed and replaced with "cell surface receptor" or "targeted receptor".  Other receptors are used so why single out PD-L1?

13) In line 383, the authors state that checkpoint therapy is not effective in most patients. A reference needs to be cited here to back up this statement. These check point inhibitors have revolutionized cancer therapy. They are not perfect, but they do have anti-cancer properties so this statement needs to be better balanced. Eliminating the cancer cells expressing PD-L1 would be beneficial (may enhance tumor antigen uptake and processing by antigen presenting cells thus further stimulating the immune response) when used in combination with checkpoint inhibitors. 

14) The authors need to stay away from directly quoting articles (see lines 389, 392, and ). The authors should paraphrase the findings of the articles cited and give more details as to why those studies are important. 

15) The statement "Using an antibody therapy reduced long-term mom;iocations from standard therapy" in lines 397-398 is ambiguous and doesn't really fit  in the paragraph. The point the authors are trying to make here is unclear. 

16) In line 419-420, the authors state that PCI is used to specifically deliver anti-CD133-SAP to targeted cancer stem cells. How is this accomplished? Is this because the light source can be given to a specific area of tissue? This needs to be clarified. 

17) The paragraph starting in line 449 discusses B-cell targeting in  Rheumatoid Arthritis. This paragraph is poorly written. This is actually a prodrug strategy cited in ref 90. The authors need to better explain the conjugate used and why it is important. 

18) Section 2.3 "Immunology" might be better suited as "Autoimmune Diseases" or something similar. Immune cells are targeted to prevent autoreactivity. In the T-cell targeting section, the authors need to be clear why T cells are being targeted in the specific situations described. As it is, it is very confusing. 

19) In line 470-473 the authors describe a study where the targeted population of cells rebounded and a higher percentage of cells (compared to before depletion) was observed. This has very important implications. Why would you want to deplete these cells using this method if they are coming back stronger than before? 

20) For what application would dendritic cells need to be depleted? also, what animal model was used for the studies described in line 476.

21) The sentence "Streptavidin-ZAP has w ice range of potential uses and is not limited to any specific field" in line 494 is out of place. This general sentence should be included in the introduction and should be followed by some examples (cancer, autoimmune diseases, etc). 

22) The two paragraphs in Section 2.5  Gastroenterology & Cardiac Function are incomplete and lacking details. The studies cited should be better described. 

Round 2

Reviewer 3 Report

The authors have addressed my previous concerns.